# The Species of the *mali*-Group of *Aphelinus* (Hymenoptera: Aphelinidae), with Descriptions of Three New Species, DNA Sequence Data and One Newly-Recorded Species from China

**DOI:** 10.3390/insects15120945

**Published:** 2024-11-29

**Authors:** Zhigang Dong, Ye Luo, Junqing Ge, Jian Huang, Andrew Polaszek, Zhuhong Wang

**Affiliations:** 1State Key Laboratory of Ecological Pest Control for Fujian and Taiwan Crops, Fujian Agriculture and Forestry University, Fuzhou 350002, China; d18235546505@163.com (Z.D.); 19957242392@163.com (Y.L.); jhuang1234@126.com (J.H.); 2Institute of Biotechnology, Fujian Academy of Agricultural Sciences, Fuzhou 350003, China; jqge@163.com; 3Science: Research, Natural History Museum, London SW7 5BD, UK

**Keywords:** Aphididae, Aphidoidea, aphid parasitoids, biocontrol, COI, taxonomy

## Abstract

*Aphelinus* species (Hymenoptera: Aphelinidae) are primary parasitoids of aphids (Hemiptera: Aphididae), playing an important role as biological control agents of aphid pests. The genus currently contains 99 described species. The *mali*-group is an important taxon within *Aphelinus*, but identification of closely related species within the group has been problematic. The *mali*-group was proposed and defined mainly based on characters of the body and leg colour, and, importantly, the fore wing with the linea calva bordered proximally with only one complete row of setae, and open posteriorly. At present, the *mali*-group consists of fourteen described species, and three species described in this paper, including eight species found in China. Here we summarize the present status of the *mali*-group, and describe three new species of the group and note one species new to China. An identification key to species of the *mali*-group is provided, excluding two species with inadequate original descriptions.

## 1. Introduction

The known species of *Aphelinus* Dalman, 1820 (Hymenoptera: Aphelinidae) are primary endoparasitoids of aphids (Hemiptera: Aphididae), with a long history of use in biological control programmes against aphid pests [1,2]. Currently *Aphelinus* contains 99 described species worldwide [3] with 31 occuring in China [2,3,4,5,6,7,8,9,10,11,12].

There are several groups of closely related species within *Aphelinus* [13,14,15]. Zehavi & Rosen (1989) [13] first proposed and defined the *mali*-group, comprising 5 species: *Aphelinus campestris* Yasnosh, 1963, *Aphelinus gossypii* Timberlake, 1924, *Aphelinus mali* (Haldeman, 1851), *Aphelinus paramali* Zehavi & Rosen, 1989 and *Aphelinus prociphili* Carver, 1980, all sharing the following characters: fore wing with a complete row of setae inside the linea calva; head and body black except for parts of metasoma pale; legs with mesofemur dark, and metafemur pale. Hayat (1998) [14] expanded the definition of the *mali*-group defined by Zehavi & Rosen (1989) [13], mainly adding that the linea calva of the fore wing is bordered proximally with one complete and 1–2 incomplete lines of setae, including the remaining species except the *abdominalis*-, *asychis*- and *nepalensis*- groups in the subgenus *Aphelinus* in India.

Hopper et al. (2012) [15] redefined the *mali*-group based on Zehavi & Rosen (1989) [13], supported by a combination of characters: head and body dark with parts of metasoma pale; fore wing with a complete row of setae inside the linea calva, or also only with a few setae in the angle between the linea and marginal vein; legs with mesocoxae, metacoxae and metatibia dark, metafemur pale. However, Hopper et al. (2012) [15] took *A. prociphili*, confirmed by Zehavi & Rosen (1989) [13], out of the *mali*-group, considering it different by having more than 1 line of setae in the delta region of the fore wing.

Following Hopper et al. (2012) [15], the *mali*-group currently consists of fourteen described species, and three species described in this paper, including eight species found in China.

In this paper, the species of the *mali*-group of *Aphelinus* worldwide and from China are reviewed, in which three new species, *Aphelinus tuberocephalus* Wang & Huang, **sp.n**., *Aphelinus hainanensis* Wang & Huang, **sp.n.** and *Aphelinus ruellia* Wang & Huang, **sp.n.** are described and illustrated, and one species, *Aphelinus coreae* Hopper & Woolley, is recorded as new to China. A key to species of the *mali*-group is provided. Moreover, based on COI we present a maximum likelihood phylogenetic tree for five sequences of four species of *Aphelinus* reported in NCBI (GenBank) and seven species in this paper, to analyze the relationships of some species in the *mali*-group.

## 2. Materials and Methods

### 2.1. Collection of Parasitoids

Aphid specimens were collected on different plants in Fujian and Hainan Provinces, and observed in the laboratory for parasitoid emergence. *Aphelinus* parasitoids, reared from these aphids, were preserved in 100% ethanol after emergence.

### 2.2. Photographs and Slides of Parasitoids

Body colour of specimens was described and photographed before slide-mounting. Specimens were mounted on slides, following the method described by Noyes (1982) [16]. Specimens were photographed using a Nikon DS-Ri2 camera, with NIS-Elements D v4.40 software, attached to a Nikon SMZ18 microscope. Slide-mounted specimens were photographed by a Nikon Ni microscope with the same camera and software. Body length was measured from specimens before being slide-mounted, and all other measurements were taken from slide-mounted specimens. Type material and the other specimens examined in the study are deposited in the College of Plant Protection, Fujian Agriculture and Forestry University, Fuzhou, China (FAFU), with some indicated specimens deposited in the Natural History Museum, London UK (NHMUK).

### 2.3. DNA Sequencing

The extraction of genomic DNA was based on the method of Polaszek et al. (2013) [17] and used the DNeasy Blood & Tissue Kit (Qiagen, Hilden, Germany), which the specimens remained intact after extraction. The specimens were subsequently preserved permanently in Canada balsam. Primer sequences and cycling conditions are given in Table 1 [18]. Polymerase chain reaction (PCR) was performed on 2 µL genomic DNA extract. Each 25 µL reaction mix contained 12.5 μL 2×Taq PCR MasterMix II, 1 µL forward primer, 1 μL reverse primer, 2 μL genomic DNA extract and 8.5 µL ddH_2_O. DNA was sequenced at Sangon Biotech (Shanghai) using the same primers used for the PCR reaction product. Forward and reverse sequences were assembled and edited using DNAMAN version 9.0 and submitted to GenBank (GenBank notes see Table 2).

### 2.4. Phylogenetic Analysis

The species and their GenBank accession numbers used in phylogenetic analysis are shown in Table 2. Based on COI the maximum likelihood phylogenetic tree was constructed by Phylosuit v1.2.3 [19]. Using the partial sequences determined in this paper and downloaded from NCBI-GenBank, the maximum likelihood phylogenetic tree was constructed with two *Coccophagus* species (Aphelinidae) as the outgroup. Firstly, MAFFT was used for sequence alignment. Secondly, ModelFinder was used to select the optimal evolution model. Finally, the maximum likelihood phylogenetic tree was constructed using IQ-TREE. The phylogenetic tree was edited in ITOL.

**Table 2 insects-15-00945-t002:** The species and GenBank accession numbers used in phylogenetic analysis.

Family/Species	GenBank Accession	Source
Aphelinidae		
*Aphelinus abdominalis*	PP480262.1	GenBank
*A. abdominalis*	PP480263.1	GenBank
*A. asychis*	OP019281.1	GenBank
*A. gossypii*	OP342775.1	GenBank
*A. varipes*	OQ789199.1	GenBank
*A. coreae* (Y18)	PQ375438.1	This paper
*A. hainanensis*, **sp.n**.	PQ375436.1	This paper
*A. humilis* (Y4)	PQ375437.1	This paper
*A. hyalopteraphidis*	PQ375442.1	This paper
*A. maidis* (Y29)	PQ375440.1	This paper
*A. ruellia*, **sp.n**.	PQ375439.1	This paper
*A. tuberocephalus*, **sp.n**.	PQ375441.1	This paper
*Coccophagus ceroplastae*	KY606141.1	GenBank
*C. japonicus*	KY605984.1	GenBank

### 2.5. Terminology and Abbreviations

Terminology follows Huang (1994) [2] with some modification, and the following abbreviations are used: F1, F2, etc. = funicle antennomeres 1, 2, etc.; T1, T2, etc. = gastral tergites 1, 2, etc.

Abbreviations for depositories as follows:
FAFUCollege of Plant Protection, Fujian Agriculture and Forestry University, Fuzhou, Fujian, China.NHMUKNatural History Museum, London, UK.TAMUTexas A&M University, College Station, USA.USNMUnited States National Museum of Natural History, Washington DC, USA.

## 3. Results

### 3.1. Modified Key Couplets to Species of the mali-Group of Aphelinus by Hopper et al. (2012) [15]

However, *Aphelinus niger* Girault, 1913 and *A*. *siphonophorae* Ashmead, 1888 were excluded from the key because of inadequate descriptions.

Couplets:

5. Pro- and metafemur dark; protibia dark or dark with yellow apex……………………..6

– Pro- and metafemur partly yellow (Figure 1g,h); protibia yellow (Figure 1h)…….6A

6. Scape yellow to pale brown with apical third yellow; pedicel and club infuscate-brown; metabasitarsus dark; F3 subquadrate in female and 1.2–2× as long as broad in male; metasoma dark with pale base………………………………………………..***A. basilicus***

– Scape dark-brown to black; pedicel yellow in female and yellow to dusky in male; club yellow; metabasitarsus yellow; F3 longer than broad in female and more than 2× as long as broad in male; metasoma dark………………………………………….***A. sanbornia***

6A. Metasoma with T1 white to yellow, other black; or T1–T3 yellow, others dark-brown to black………………………………………………………………………………………..7

– Metasoma with T1 yellow, T2–T4 black brown, except the anterior part of T2 yellow, T5–T7 yellow (Figure 1f)…………………………………………….***A. tuberocephalus*, sp.n.**

Couplets:

11. Club more than 3× as long as broad in male; mesotibia dark with base and apex pale………………………………………………………………………………….***A. rhamni***

– Club less than 3× as long as broad in male (Figures 4c, 6c and 8c); mesotibia dark with distal half yellow (Figures 3g, 5i and 7h)……………..12

12. Mid-lobe of mesoscutum with less than 40 setae (Figure 3b)……***A. hainanensis***, **sp.n.**

– Mid-lobe of mesoscutum with more than 40 setae (Figures 5g and 7e)………..13

13. Scape white and base black in female (Figure 5c); scape black in male (Figure 6c)………………………………………………………………………………***A. ruellia***, **sp.n.**

– Scape yellow in female (Figure 7c); scape dark yellow-grey in male (Figure 8c)……………………………………………………………………………………..***A. coreae***

### 3.2. Species Accounts

#### 3.2.1. *Aphelinus tuberocephalus* Wang & Huang, **sp.n.** (Figure 1 and Figure 2)


https://zoobank.org/43AC8B42-C78B-40ED-9E97-3EE7FAD20B8F


*Diagnosis*. *Aphelinus tuberocephalus*, **sp.n.** resembles *A. glycinis*, but can be distinguished from it by: forecoxae black (*A. glycinis* yellow-white); mesofemur mostly black, the base pale yellow (*A. glycinis* yellow-white); mesotibia mostly black, the apex pale yellow (*A. glycinis* yellow-white, the middle grey); tarsus pale yellow except hind leg basitarsus black-brown (*A. glycinis* pale brown, the apex and hind leg basitarsus grey-brown); T1 yellow, T2–T4 black-brown, except the anterior part of T2 yellow, T5–T7 yellow (*A. glycinis* T1&T2, the apex and sternum of metasoma yellow, others brown).

*Aphelinus tuberocephalus*, **sp.n.** also resembles *A. coreae*, but can be separated from it by: forefemur pale yellow (*A. coreae* dark grey, the distal half pale); mesofemur mostly black, the base pale yellow (*A. coreae* dark grey to black); mesotibia mostly black, the apex pale yellow (*A. coreae* dark-grey to black, the distal half pale); T1 yellow, T2–T4 black-brown, except the anterior part of T2 yellow, T5–T7 yellow (*A.coreae* metasoma dark-brown, the base and apex yellow).

*Description*. **Female**. Body length 0.82–1.12 mm (Holotype 1.02 mm). **Colour**. Head black; eyes red-brown; antennae yellow-brown; mesosoma black-brown; legs with coxae black, fore leg with trochanter, femur, tibia, tarsus white to pale yellow, mid leg with trochanter white to pale yellow, mesofemur and mesotibia mostly black, except the base of mesofemur and the apex of mesotibia white to pale yellow, mesotarsus pale yellow, metafemur white to pale yellow, metatibia black expect the base white, the basitarsus of hind leg black, others pale yellow; claw black. T1 yellow, T2–T4 black-brown, except the anterior part of T2 yellow, T5–T7 yellow. The third valvula white. **Head**. Head 1.23× as broad as high in frontal view, about as broad as mesosoma; frontovertex 0.37× head width and about as broad as scape length; mandible with two acute teeth; antennae with scape 3.60× as long as broad, pedicel 1.39× as long as broad, F1 nearly rectangular, 1.43× as broad as long, F2 1.21× as broad as long, F3 trapezoid, club 2.25× as long as broad and 2.90× longer than F3. **Mesosoma**. Mesonotum with fine reticulate sculpture, mid-lobe of mesoscutum with 30–50 short setae and 2 pairs of long setae, side lobes each with 1 long and 1 short setae; mesoscutellum with 2 pairs of long setae. **Fore wing**. 2.28× as long as broad; costal cell with 3 rows of setae, costal cell 1.41× as long as marginal vein, submarginal vein with 5 setae; marginal vein with 8 setae along anterior margin; stigmal vein short; delta region proximal to linea calva with one complete line of 13 setae and 2 additional setae in angle with marginal vein; wing disc with dense setae. **Metasoma**. 1.76× as long as mesosoma; ovipositor located basally at T2, 1.96× as long as mesotibia, third valvula 0.40× length of ovipositor.

**Male**. Body length 0.70–1.01 mm. Similar to female except: antenna with scape swollen medially, 3.89× as long as broad, maximum width 2.68× distal end width, with 3 crater-shaped secretory pores in line on ventral surface; pedicel 1.72× as long as broad; F1, F2 trapezoid, F3 rectangle and 0.89× as long as broad; club 2.82× as long as broad. Metasoma 1.23× as long as mesosoma. Antenna with scape and pedicel brown yellow, the rest light yellow; fore leg with femur black basally, the rest white; T1 yellow, the rest of metasoma black.

Host. *Tuberocephalus momonis* (Matsumura) (Hemiptera: Aphididae) on *Prunus persica* (L.) Batsch (Rosales: Rosaceae).

*Distribution*. China (Fujian).

*Etymology*. The new species was named after the genus of the host aphid, *Tuberocephalus*.

Material. **Holotype** ♀, No. D10, **ex.**
*Tuberocephalus momonis* (Matsumura) on *Prunus persica* (L.) Batsch. China: Fujian, Sanming, Youxi, Yangzhong FAFU Science and Education Base, 2.vi.2022, coll. Zhuhong Wang (FAFU).
Figure 1*Aphelinus tuberocephalus*, **sp.n.** female. (**a**) adult in dorsal view; (**b**) adult in lateral view; (**c**) mesosoma; (**d**) antenna; (**e**) fore wing; (**f**) metasoma; (**g**) mid leg; (**h**) fore leg; (**i**) hind leg.
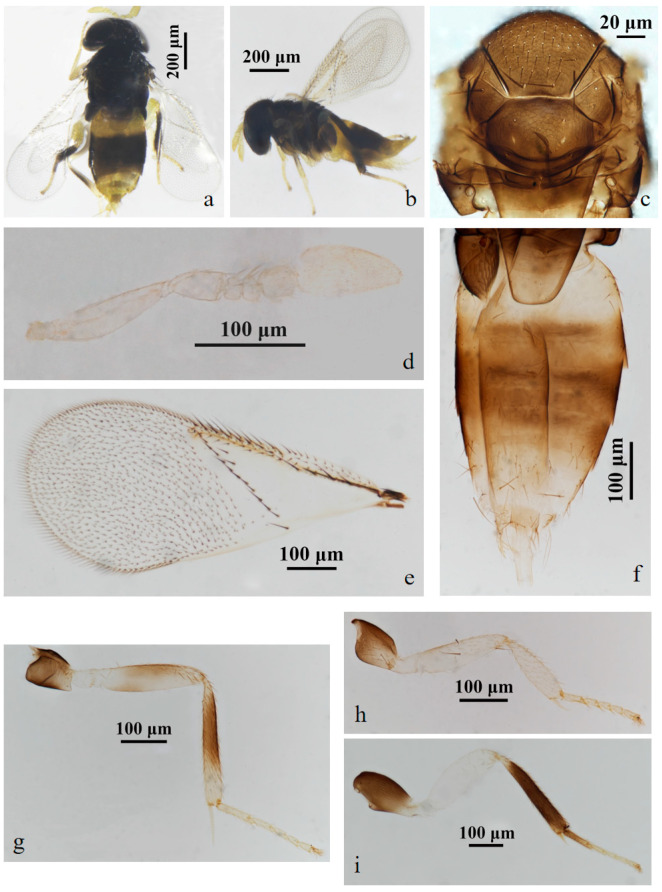

Figure 2*Aphelinus tuberocephalus*, **sp.n.** male. (**a**) adult in dorsal view; (**b**) adult in lateral view; (**c**) mesosoma; (**d**) antenna; (**e**) fore wing; (**f**) metasoma; (**g**) mid leg; (**h**) fore leg; (**i**) hind leg.
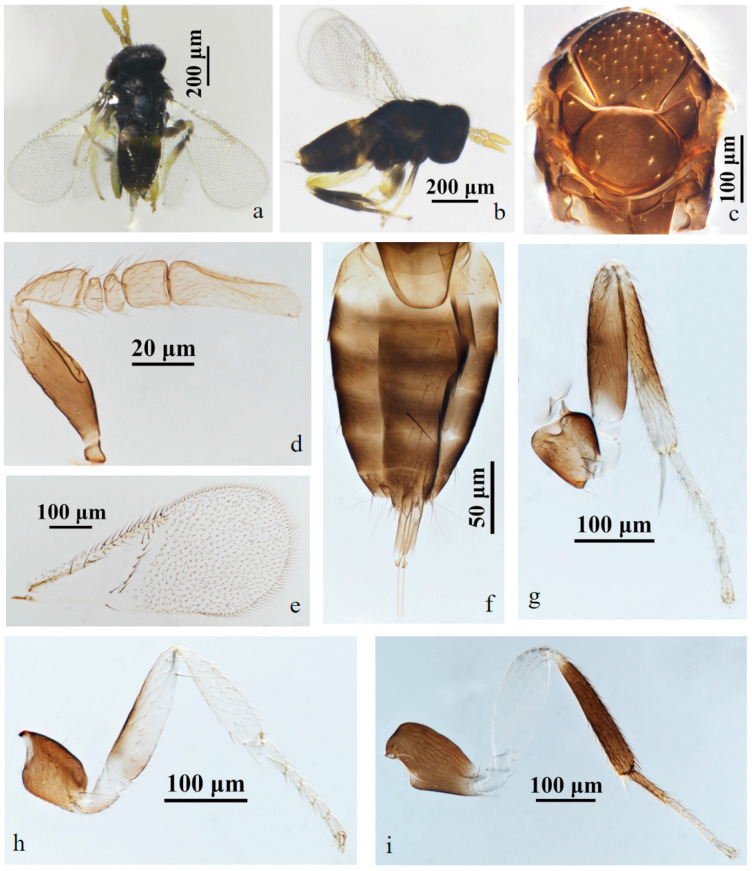


#### 3.2.2. *Aphelinus hainanensis* Wang & Huang, **sp.n.** (Figure 3 and Figure 4)


https://zoobank.org/4DCFAECE-BAC2-40DA-9936-21CC0027F34D


*Diagnosis*. *Aphelinus hainanensis*, **sp.n.** resembles *Aphelinus rhamni* Hopper & Woolley but can be distinguished from it by: antennae with the basal 1/2 of scape grey, grey-white apically, pedicel, funicle and club yellow-brown (*A. rhamni* antennae yellow, sometimes the basal 1/2 of scape and pedicel grey-yellow); the basal 1/2 of mesotibia black, yellow-brown apically (*A. rhamni* mesotibia dark-grey to black, and the base and apex pale); T1 yellow, T2–T7 dark-brown (*A. rhamni* T1–T7 yellow-brown, the base and apex yellow); fore wing with one complete line of 12 setae, triangle area without setae, linea calva open posteriorly (*A. rhamni* with one complete line of 13–15 setae, and the triangle area with 2–6 setae, linea calva semi-closed posteriorly).

*Description*. **Female**. Body length 0.85–1.00 mm (Holotype 0.97 mm). **Colour**. Body black; head black; eyes dark brown; antennae with the basal 1/2 of scape grey, grey-white apically, pedicel, funicle and club yellow-brown; mesonotum black; legs with coxae black, the basal 1/2 of forefemur black, the apex of forefemuer, foretibia and foretarsus pale yellow-brown, mesofemur black, the basal 1/2 of mesotibia black, the apex of mesotibia and mesotarsus pale yellow-brown, metafemur white, metatibia and basitarsus black, claw black; T1 yellow, T2–T7 dark-brown; peripheral third valvula black, the rest white. **Head**. Head 1.15× as broad as high in frontal view, about as broad as mesosoma; frontovertex 0.45× head width and 0.78× as broad as scape length; mandible with two acute teeth; antennae with scape 5.14× as long as broad, pedicel 1.58× as long as broad, F1, F2 nearly elliptical, F1 1.50× as broad as long, F2 1.87× as broad as long, F3 nearly rectangular, 1.10× as broad as long, club 2.90× as long as broad and 3.49× as long as F3. **Mesosoma**. Mesonotum with fine reticulate sculpture, mid-lobe of mesoscutum with about 25–30 short setae and 2 pairs of long setae, side lobes each with 1 long and 1 short seta; mesoscutellum with 2 pairs of long setae. **Fore wing**. 2.27× as long as broad; costal cell with 3 rows of setae, costal cell 1.27× as long as marginal vein; submarginal vein with 5 setae, marginal vein with 6 setae along anterior margin, stigmal vein short; delta region proximal to linea calva with one complete line of 12 setae; wing disc with dense setae. **Metasoma**. 1.39× as long as mesosoma; ovipositor located basally at T2, 1.35× as long as mesotibia, third valvula 0.30× length of ovipositor.

**Male.** Body length 0.62–0.87 mm. Similar to female except: antenna with scape swollen medially, 3.19× as long as broad, maximum width 2.48× distal end width, with 3 crater-shaped secretory pores in line on ventral surface; pedicel 1.77× as long as broad; F1 0.70× as long as broad, F2 trapezoid, F3 square; club 2.81× as long as broad. Metasoma 1.25× as long as mesosoma. Antenna with the basal 2/3 of scape and pedicel brown-yellow, funicle slightly smoky-brown, the rest light yellow; fore leg with the base 2/3 of femur black, the rest white; T1 yellow, the rest of metasoma black.

Host. Aphid on *Hibiscus rosa-sinensis* L. (Malvales: Malvaceae).

*Distribution*. China (Hainan).

*Etymology*. The new species was named after the collection locality, Hainan.

Material. **Holotype** ♀, No. H1, **ex. aphid** on *Hibiscus rosa-sinensis* L. China: Hainan, Wenchang, Dongjiao, Qinglan Bridge Bridgehead Park, 30.iii.2023, coll. Zhigang Dong (FAFU).
Figure 3*Aphelinus hainanensis*, **sp.n.** female. (**a**) adult in lateral view; (**b**) mesosoma; (**c**) fore wing; (**d**) antenna; (**e**) metasoma; (**f**) fore leg; (**g**) mid leg; (**h**) hind leg.
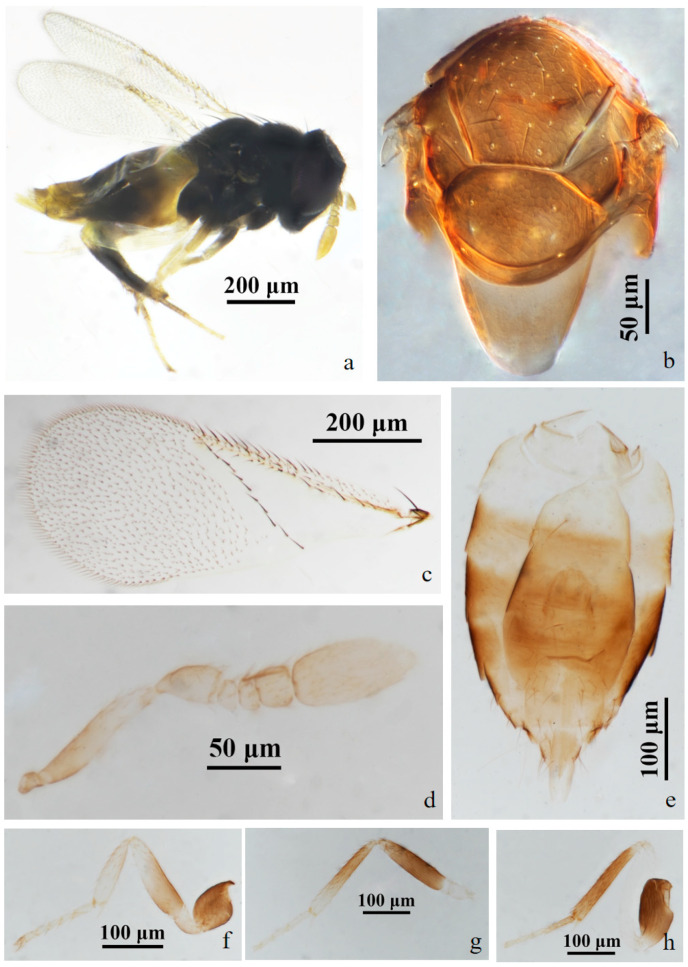

Figure 4*Aphelinus hainanensis*, **sp.n.** male. (**a**) fore wing; (**b**) mesosoma; (**c**) antenna; (**d**) fore leg; (**e**) metasoma; (**f**) mid leg; (**g**) hind leg.
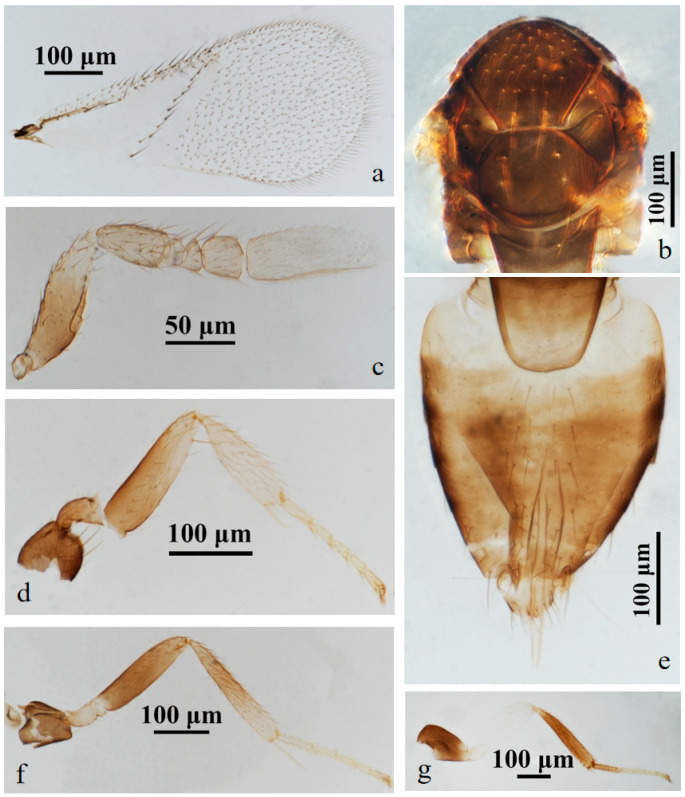


#### 3.2.3. *Aphelinus ruellia* Wang & Huang**, sp.n.** (Figure 5 and Figure 6)


https://zoobank.org/63D5F103-1B5E-4DF4-ABB6-0BAB2EAFF1D8


*Diagnosis*. *Aphelinus ruellia*, **sp.n.** resembles *Aphelinus coreae* Hopper & Woolley, 2012 [15], but can be separated from it by: antennae with scape mostly white, yellow-brown basally, 5.2× as long as broad, other segments yellow-brown, club 2.4× as long as broad (*A. coreae* antennae yellow, scape 4.0× as long as broad, club 3.75× as long as broad); metasoma dark-brown, T1 yellow, the apex of metasoma dark black-brown (*A. coreae* metasoma dark-brown, the base and apex yellow). Antennae in male with scape black, the sensilla narrow and long with 4 crater-shaped secretory pores in line on ventral surface, pedicel and funicel grey-black, club yellow-brown (*A. coreae* antennae with scape dark yellow-grey, the apex 1/2 of scape yellow, with 2–3 crater-shaped secretory pores in line on ventral surface, pedicel grey-yellow).

*Description*. **Female**. Body length 0.68–0.94 mm. **Colour**. Body black; antennae with scape mostly white, the base yellow-brown, other segments yellow-brown; legs with coxae black; the basal 1/2 of forefemur black, the apex white, foretibia and foretarsus pale yellow-brown; mesofemur white basally, others black, mesotibia white apically, others black, mesotarsus pale yellow-brown; metafemur white, metatibia white basally, others black, basitarsus black and other metatarsus yellow-brown; claw black; T1 yellow, metasoma dark yellow-brown apically; the peripheral of third valvula black, others white. **Head**. Head 1.23× as broad as high in frontal view, about as broad as mesosoma; frontovertex 0.41× head width; mandible with two acute teeth; antennae with scape 5.20× as long as broad, pedicel 1.80× as long as broad, F1 1.44× as broad as long, F2 1.62× as broad as long, F3 subquadrate, club 2.40× as long as broad. **Mesosoma**. Mesonotum with fine reticulate sculpture, mid-lobe of mesoscutum with about 50–55 short setae and 2 pairs of long setae, side lobes each with 1 long and 1 short setae; mesoscutellum with 2 pairs of long setae; mesotibial spur 0.96× mesobasitarsus. **Fore wing**. 2.26× as long as broad; costal cell with 3 rows of setae, costal cell 1.17× as long as marginal vein; submarginal vein bearing 5 setae, marginal vein bearing 7–8 setae along anterior margin, stigmal vein short; delta region proximal to linea calva with one complete line of 12 setae, trigonum area 3 setae; wing disc with dense setae. **Metasoma**. 1.32× as long as mesosoma; ovipositor located basally at basal 0.5× of metasoma, 1.15× as long as mesotibia, third valvula 0.52× length of ovipositor.

**Male**. Body length 0.58–0.83 mm. Similar to female except: antenna with scape swollen medially, 3.02× as long as broad, maximum width 2.34× distal end width, with 4 crater-shaped secretory pores in line on ventral surface; pedicel 1.86× as long as broad; F1 trapezoid, F2 annular, F3 rectangle; club 2.82× as long as broad. Abdomen 1.45× as long as mesosoma. Antennal scape black.

*Host*. Aphid on *Ruellia simplex* C.Wright (Lamiales: Acanthaceae).

*Distribution*. China (Fujian).

*Etymology*. The new species was named after the genus of the host plant, *Ruellia*.

Material. **Holotype** ♀, No.Y19, **ex. aphid** on *Ruellia simplex* C.Wright. China: Fujian, Fuzhou, Gulou, Zuohai Park, 16.v.2023, coll. Zhigang Dong and Ye Luo (FAFU).
Figure 5*Aphelinus ruellia*, **sp.n.** female. (**a**) adult in dorsal view; (**b**) adult in lateral view; (**c**) antenna; (**d**) fore wing; (**e**) hind leg; (**f**) metasoma; (**g**) mesosoma; (**h**) fore leg; (**i**) mid leg.
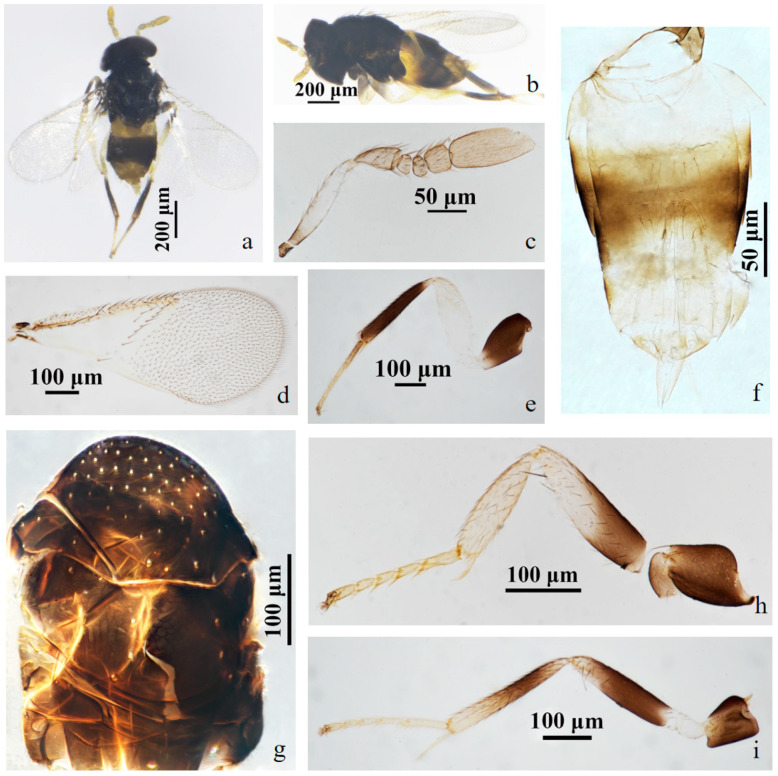

Figure 6*Aphelinus ruellia*, **sp.n.** male. (**a**) adult in dorsal view; (**b**) adult in lateral view; (**c**) antenna; (**d**) fore wing; (**e**) mesosoma; (**f**) metasoma; (**g**) mid leg; (**h**) hind leg; (**i**) fore leg.
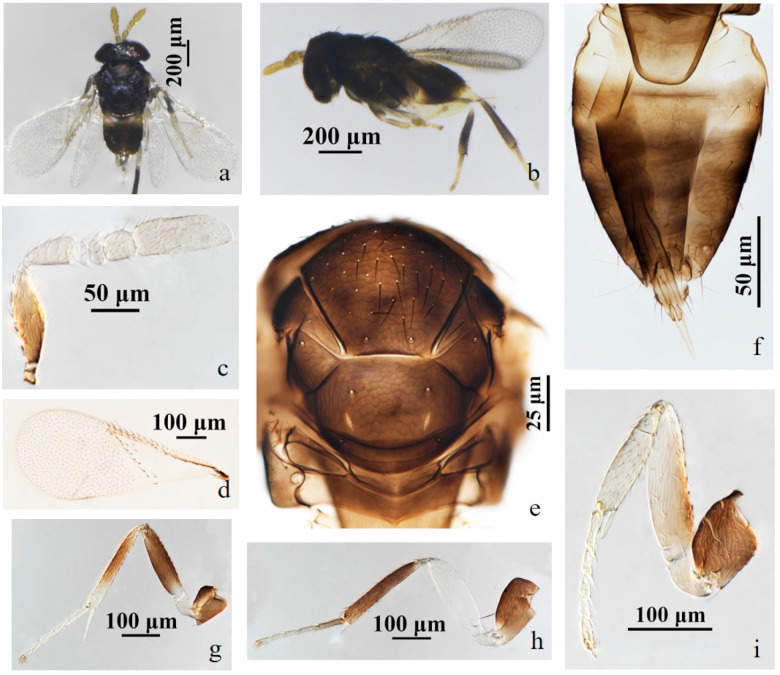


#### 3.2.4. *Aphelinus coreae* Hopper & Woolley, 2012 [15] (Figure 7 and Figure 8)

*Aphelinus coreae* Hopper *&* Woolley, 2012, in: Hopper et al. [15], Journal of Hymenoptera Research, 26: 90. **Holotype** ♀, Paratypes 16♀14♂, Korea: Gyeongsangnam, Miryang. ex. *Aphis glycines* on soybean (USNM, TAMU, NHMUK).

*Host*. *Aphis glycines* Matsumura (Hemiptera: Aphididae) on soybean [15].

*Distribution*. Korea (Gyeongsangnam), China (Fujian, Hainan) new country record.

Material. 1♀1♂, No.Y20, **ex. aphid** on *Nerium oleander* L. China: Fujian, Fuzhou, Gulou, 27.v.2023, coll. Zhigang Dong; 1♀, No.Y21, **ex. aphid** on *Hibiscus mutabilis* L. China: Fujian, Fuzhou, Minhou, Qishan Lake Park, 23.v.2023, coll. Ye Luo; 1♀, No.Y3, **ex. aphid** on *Ilex cornuta* Lindl. & Paxton. China: Fujian, Fuzhou, Cangshan, 11.iv.2023, coll. Zhigang Dong; 1♀, No.Y18, **ex. aphid** on *Nerium oleander* L. China: Fujian, Xiamen, Egret Chau Park, 25.v.2023, coll. Zhigang Dong; 1♀, No.Y41, China: Hainan, Sanya, 10.i.2014, coll. Zhuhong Wang; 1♀, No.Y15, **ex. aphid** on *Paederia foetida* L. China: Fujian, Fuzhou, Minhou, 17.v.2023, coll. Zhigang Dong; 1♀, No.Y15, **ex. aphid** on *Paederia foetida* L. China: Fujian, Fuzhou, Minhou, 18.x.2023, coll. Zhigang Dong and Ye Luo; 1♀, No.W15, **ex. aphid** on *Camellia sinensis* (L.) O.Ktze. China: Fujian, Wuyishan, 15.vii.2023, coll. Zhigang Dong (FAFU; NMHUK).
Figure 7*Aphelinus coreae* Hopper *&* Woolley. female. (**a**) adult in dorsal view; (**b**) adult in lateral view; (**c**) antenna; (**d**) fore wing; (**e**) mesosoma; (**f**) metasoma; (**g**) fore leg; (**h**) mid leg; (**i**) hind leg.
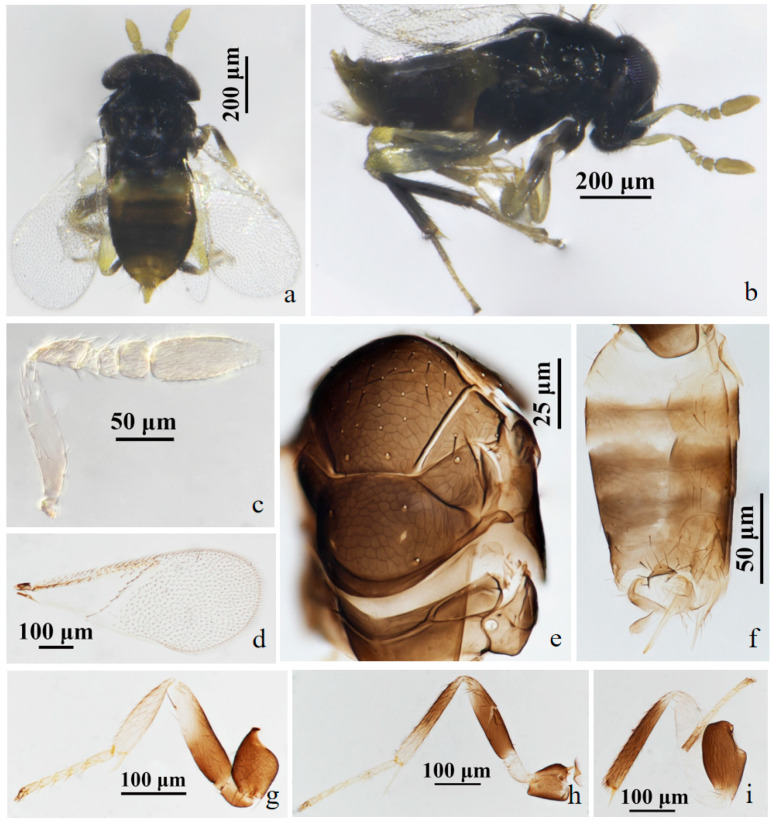

Figure 8*Aphelinus coreae* Hopper *&* Woolley. male. (**a**) adult in dorsal view; (**b**) adult in lateral view; (**c**) antenna; (**d**) fore wing; (**e**) mesosoma; (**f**) metasoma; (**g**) fore leg; (**h**) hind leg; (**i**) mid leg.
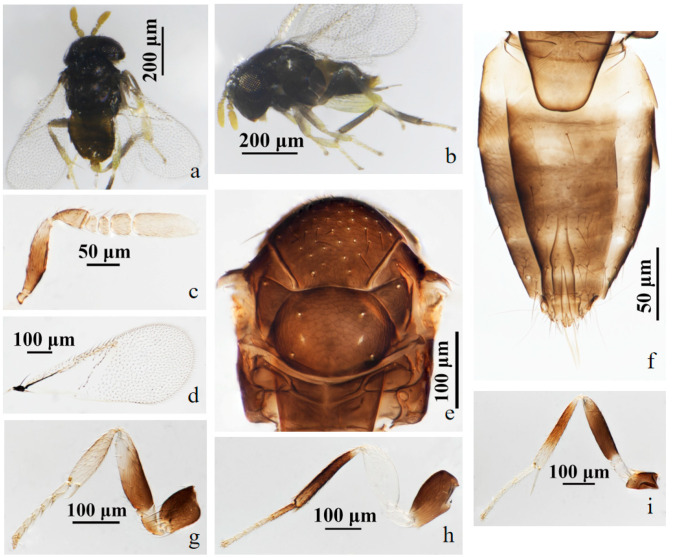


### 3.3. Phylogenetic Analysis

The phylogenetic relationships among some species of *Aphelinus* were investigated and the results are shown in Figure 9. The species of *Aphelinus* clustered on the same branch, and the species of the *mali*-group (*A. gossypii*, *A. tuberocephalus*, **sp.n.**, *A. ruellia*, *A. coreae* and *A. hainanenesis*, **sp.n.**) also clustered on the same branch which formed a sister group together with (*A. varipes* + (*A. maidis* + *A. hyalopteraphidis*)) + (*A. asychis* + *A. abdominalis*). In addition, the *mali*-group and (*A. varipes* + (*A. maidis* + *A. hyalopteraphidis*)) + (*A. asychis* + *A.abdominalis*) formed a sister group together with *A. humilis*. Moreover, *A. ruellia*, **sp.n.** formed a sister group together with *A. coreae*, which is consistent with the morphological classification of the two species.

## 4. Discussion

The *mali*-group of *Aphelinus* was first proposed and defined by Zehavi & Rosen (1989) [13], comprising 5 species, based on body and leg colour, and fore wing with only one complete row of setae bordered proximally on linea calva. Hayat (1998) [14] expanded the definition of the *mali*-group defined by Zehavi & Rosen (1989) [13], mainly as linea calva of fore wing bordered proximally with one complete and 1–2 incomplete lines of setae, including more species of the subgenus *Aphelinus* except the species of *abdominalis*-, *asychis*- and *nepalensis-* groups in the subgenus in India. Hopper et al. (2012) [15] redefined the *mali*-group based on Zehavi & Rosen (1989) [13], supported by a combination of characters, in which the main character is a single complete row of setae proximal to the linea calva of fore wing, or with a few additional setae in the angle between this row and the marginal vein. In this paper, following the work of Hopper et al. (2012) [15], the three new species of the *mali*-group were described. At present, the *mali*-group of *Aphelinus* consists of seventeen species worldwide and including eight species distributed in China (Table 3).

The relationships among populations in the *varipes*-group of *Aphelinus* were analyzed by Heraty et al. (2007) [27], using four nuclear (28S-D2, ITS1, ITS2, ArgK) and two mitochondrial (COI, COII) gene regions, in order to infer the molecular phylogenetics and reproductive incompatibility in the *varipes*-group. Gokhman et al. (2017) [28] measured and determined the genome sizes and karyotypes of some species groups of *Aphelinus*, showing that large differences in genome size and karyotype were among the *mali*-, *daucicola-* and *varipes*-groups. However, the differences in genome size and total chromosome length between species groups matched the phylogenetic divergence between them. Up to now, there are few molecular reports on the species groups of *Aphelinus*. This paper provides preliminarily support for the taxonomic and molecular relationships among the some species of the *mali*-group. However, the COI gene sequence of all species of this group and other group species of *Aphelinus* has not been obtained. Therefore, the corresponding comprehensive gene sequence should be obtained in the future to more accurately investigate the phylogenetic relationships among the species of various groups of *Aphelinus*.

## Figures and Tables

**Figure 9 insects-15-00945-f009:**
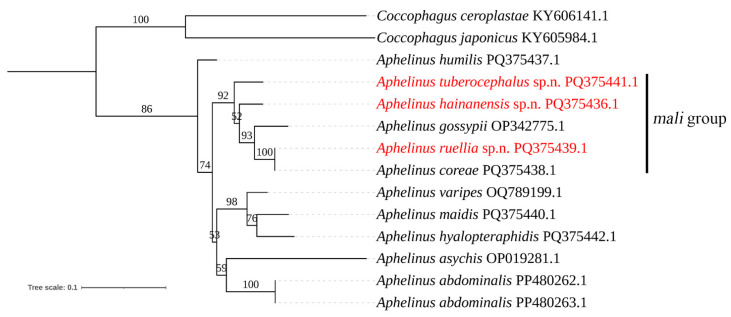
The maximum likelihood phylogenetic tree based on the *Aphelinus* COI gene. Bootstrap support values indicated on branches.

**Table 1 insects-15-00945-t001:** Primers and cycling conditions.

Primer	Sequence	Cycling Conditions			
LCO1490	5′-GGTCAACAAAATCATAAAGATATTGG-3′	Denaturation	Annealing	Extension	Cycles
HCO2198	5′-TAAACTTCAGGGTGACCAAAAAATCA-3′	94 °C (30 s)	47 °C (30 s)	72 °C (1 min)	35

**Table 3 insects-15-00945-t003:** The described species of the *mali*-group of *Aphelinus*.

Species	Distribution	Hosts	References
*A. basilicus*	India, Sri Lanka, Bangladesh	aphids on *Ocimum basilicum*, *Lantana camara* and okra	[14]
*A. campestris*	Russia	aphid on *Polygonum* sp.	[13,20]
*A. coreae*	Korea, China (Fujian)	*Aphis glycines*	[15], Note in this paper
*A. engaeus*	South Africa	*Rhodesaclerda* sp. on *Loranthus dregei*	[15,21]
*A. ficusae*	South Africa	aphid on *Ficus sycomorus*	[15,21]
*A. glycinis*	China (Liaoning)	*Aphis glycines*	[15]
*A. gossypii*	USA (Honolulu); cosmopolitan in Old World tropics	aphid on *Ficus sycomorus*, *Aphis* sp. on *Hibiscus*	[13,14,15,22]
*A. hainanensis*, **sp.n.**	China (Hainan)	an unidentified aphid on *Hibiscus rosa-sinensis*	Described in this paper
*A. mali*	China, India, Cosmopolitan	*Aphis gossypii*, *A. spiraecola*, *Eriosoma lanigerum* on apple, *Myzus persicae*, ? *Aspidiotus destructor*	[2,13,14,15]
*A. niger*	Australia (Brisbane, Queensland)	unknown	[15,23]
*A. paramali*	Israel	melon aphid, *Aphis gossypii*, pomegranate aphid, *Aphis punicae*	[13,15]
*A. rhamni*	China (Beijing)	*Aphis glycines*	[15]
*A. ruellia*, **sp.n.**	China (Fujian)	an unidentified aphid on *Ruellia simplex*	Described in this paper
*A. sanborniae*	USA (Pennsylvania)	*Sanbornia juniperi*	[15,24]
*A. siphonophorae*	North America	*Siphonophora sp.*	[15,25]
*A. spiraecolae*	China (Guangdong)	*Aphis spiraecola*	[15,26]
*A. tuberocephalus*, **sp.n.**	China (Fujian)	*Tuberocephalus momonis*	Described in this paper

Note: Revised from Hopper et al. (2012) [15]. However, distribution and hosts of species of the *mali*-group have been added to the table.

## Data Availability

Data are contained within the article.

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
