# Peer review of "The Species of the mali-Group of Aphelinus (Hymenoptera: Aphelinidae), with Descriptions of Three New Species, DNA Sequence Data and One Newly-Recorded Species from China"

_insects, 2024, doi:10.3390/insects15120945_

Round 1
Reviewer 1 Report
Comments and Suggestions for Authors
Review of Dong et al. “Species of the mali-group of Aphelinus …”
This paper is potentially a valuable contribution to the systematics of the difficult genus Aphelinus. However, before it can be published, a number of issues need to be corrected or addressed.
1- The list of Aphelinus spp. in the mali group that is presented is largely consistent with my own. The authors have clearly done a thorough investigation of the group. We have also included A. meridionalisKalina 1977 in the mali group, based on the original description. I would urge the authors to have a look at this.
2- The key to species is potentially a very useful contribution. Unfortunately, the decision to right justify this text makes it extremely difficult to review it.
3- The copious illustrations range in quality from adequate to very poor, by contemporary standards. In particular, the following figures are so poor as to be essentially useless to the reader: 1b, 1f, 2a, 2b, 2c, 2i, 3a, 3b, 4a, 4b, 4c, 5a, 5b, 5f, 5g, 6a, 6b, 6c, 6d, 7a, 7b, 7e, 7f, 8a, 8b, 8c, 8i.
4- Based on the results of Heraty et al. (2007), one would not expect mtDNA region COI to provide adequate resolution of species in the mali group. Indeed, although there is reasonable support for the monophyly of the mali group in Figure 9, there is weak or no support for relationships within it. Specifically, there is only weak support distinguishing two new species, tuberocephalus and hainanensis and no differentiation whatever between the new species ruellia and A. coreae. Therefore, I looked carefully at the morphological characters that the authors used to diagnose the new species, and I believe that they are sufficient, given the difficulty of distinguishing cryptic species of Aphelinus.
5- Regarding Figure 9, I have several questions. What do the branch lengths represent (“Tree scale” is completely uninformative). Are the numbers subtending nodes bootstrap percentages? These should be spelled out in the caption to Fig. 9. Along these lines, it is unclear in the methods (lines 106-112) whether Phylosuit or IQ-Tree (or both) were used to produce the phylogeny in Fig. 9. This should also be explained in the caption.
6- The ms. Is full of typos and misspelled words, and needs to be carefully proof-read by someone.
7- Line 13, there is a superfluous sentence fragment
8-Line 21, “Where” should be replace by “here”
9- Lines 51-52, what “remaining species”?
10- line 55, delete “and”
11- line 60, what does “confirmed” mean in this context?
12- line 212, which tarsus, the basitarsus?
13- line 225, replace “expect” with “except”
14- line 250, Tuberocephalus momonis should be in italics
15- line 242, delete “name”
16- Aphelinus is consistently misspelled as “Apelinus” throughout the ms., including figure captions.
17- Scape is consistently misspelled as “scale” throughout the ms.
18- line 279, what is meant by “peripheral valvulae”, the exerted portion?
19- line 319, replace “funicle” with “funicle”
20- line 331, does “nearly squarer” (which does not make sense) mean subquadrate?
21- line 396, delete “of”
Reviewer 2 Report
Comments and Suggestions for Authors
The work is all good but needs a little revision.
Suggestions and Corrections
1. Line 33: Abstract: A. coreae Hopper & Woolley “change to” Aphelinus coreae Hopper & Woolley, 2012”. The first citation is in the full abstract.
2. Line 36: Please do not remove words from the title to compose the keywords, this may cause indexing problems for the article. Modify the keyword: Aphididae.
3. Line 41: Change “Aphelinus” by “Aphelinus Dalman, 1820”. She was cited for the first time in the text.
4. Lines 44 and 46: Include the year of description and their full names in the species name. They were mentioned for the first time in the text.
5. Lines 74 and 75: How was the methodology for collecting insects carried out, the collection period, and how many specimens were collected?
6. Line 102: Table 1. Insert more information in table 1, as the tables and figures are self-explanatory.
7 - 2.5. line 115: Terminology and Abbreviations /Suggestion: PCR, IQ–TREE, ITOL, and NCBI-GenBank.
8. Line 208: 3.2.1. Aphelinus tuberocephalus Wang & Huang, sp.n. (Figures 1,2) - Insert italics: Aphelinus tuberocephalus.
9. Line 250: Host. Tuberocephalus momonis Matsumura “ change to” Tuberocephalus momonis (Matsumura, 1917), insert italics and Order and Family.
10. 252 and 253: Tuberocephalus - Insert author, year, Order, and Family.
11-Line 254: Material. Holotype ♀, No. D10, ex. aphid on Prunus persica (L.) Batsch: Enter Year, Order, and Family.
12. Line 263: 3.2.2. Apelinus hainanensis Wang & Huang, sp.n. (Figures 3,4). Insert italics.
13. Line 264: A. rhamni, full name of the species and insert author and year - First cited in the text.
14. Line 300 and 303: Hibiscus rosa-sinensis L. Insert year, Order, Family, Insert italics, and remove the line.
15. Line 312: 3.2.3. Apelinus ruellia Wang & Huang, sp.n. - Insert italics.
16. Line 13 and Line 25: Aphelinus (Hymenoptera: Aphelinidae) - Insert year and author
17. Line 313: A. coreae - Insert author and year - First cited in the text.
18. Line 346: Host. Aphid on Ruellia simplex Leonard. Insert year, Order, and Family.
19. 349: ex. aphid on Ruellia simplex Leonard. China: Fujian “modify” ex. aphid on R. simplex Leonard. China: Fujian. Previously cited.
20. Line 357: 3.2.4. Aphelinus coreae Hopper & Woolley, 2012 (Figures 7,8) - Insert italics.
21. Host. Aphis glycines on soybean (Hopper & Woolley, 2012). Insert year, Order, and Family.
22. Line 362: Distribution. Korea (Gyeongsangnam), China (Fujian, Hainan) new country record. Replace the "comma" with "and".
23. Line 364: Hibiscus mutabilis L. . Insert year, Order, Family.
24. Figure 375: Figures 7 and 8: Apelinus coreae Hopper & Woolley – insert year.
25. Lines 365 to 372: hosts: Insert year, Order, Family.
26. Line 409: Insert Table 3 in the Results.
27. Código Internacional De Nomenclatura Zoológica – ICZN.
28. Add to Materials and Methods: What methodology was used to collect the specimens, the period of specimen collection, and the total number of individuals?
29. Why were females not described?
Comments on the Quality of English Language
The quality of English is adequate for the manuscript.
Reviewer 3 Report
Comments and Suggestions for Authors
This paper describes new species of the genus Aphelinus and reviews this genus using also genetic analyses, although some data are incomplete. I suggest to improve the manuscript adding information. Some comments are in PDF attached.
